# Bioactive Multilayer Polylactide Films with Controlled Release Capacity of Gallic Acid Accomplished by Incorporating Electrospun Nanostructured Coatings and Interlayers

**Luis Quiles-Carrillo [1]**, **Nestor Montanes [1]**, **José M. Lagaron [2]**, **Rafael Balart [1]**
and **Sergio Torres-Giner [2,*]**

1 Technological Institute of Materials (ITM), Universitat Politècnica de València (UPV), Plaza Ferrándiz y Carbonell 1, 03801 Alcoy, Spain; luiquic1@epsa.upv.es (L.Q.-C.); nesmonmu@upvnet.upv.es (N.M.); rbalart@mcm.upv.es (R.B.)
2 Novel Materials and Nanotechnology Group, Institute of Agrochemistry and Food Technology (IATA), Spanish National Research Council (CSIC), Calle Catedrático Agustín Escardino Benlloch 7, 46980 Paterna, Valencia, Spain; lagaron@iata.csic.es
* Correspondence: storresginer@iata.csic.es; Tel.: +34-963-900-022



**Featured Application:** **The electrospinning coating technology is herein proposed to develop multilayer polylactide (PLA) films with controlled release rates of gallic acid (GA), a natural antioxidant, into saline medium. The bioactivity of the PLA films described here is based on their potential capacity to incorporate a large amount of GA into the packaging structure until its eventual release into the food product either through fast release, by means of coatings, or sustained release, in the form of interlayers, during storage. Therefore, their application in bioactive packaging for both short- and long-term storage of food products can advantageously permit the development of functional food, enhancing the food impact over the consumer's health.**

**Abstract:** The present research reports on the development of bi- and multilayer polylactide (PLA) films by the incorporation of electrospun nanostructured PLA coatings and interlayers containing the antioxidant gallic acid (GA) at 40 wt% onto cast-extruded PLA films. To achieve the bilayer structures, submicron GA-loaded PLA fibers were applied on 200-μm cast PLA films in the form of coatings by electrospinning for 1, 2, and 3 h. For the multilayers, the cast PLA films were first coated on one side by electrospinning, then sandwiched with 10-μm PLA film on the other side, and the resultant whole structure was finally thermally post-treated at 150 °C without pressure. Whereas the bilayer PLA films easily delaminated and lacked transparency, the multilayers showed sufficient adhesion between layers and high transparency for deposition times during electrospinning of up to 2 h. The incorporation of GA positively contributed to delaying the thermal degradation of PLA for approximately 10 °C, as all films were thermally stable up to 345 °C. The in vitro release studies performed in saline medium indicated that the GA released from the bilayer PLA films rapidly increased during the first 5 h of immersion while it stabilized after 45–250 h. Interestingly, the PLA multilayers offered a high sustained release of GA, having the capacity to deliver the bioactive for over 1000 h. In addition, in the whole tested period, the GA released from the PLA films retained most of its antioxidant functionality. Thus, during the first days, the bilayer PLA films can perform as potent vehicles to deliver GA while the multilayer PLA films are able to show a sustained release of the natural antioxidant for extended periods.

**Keywords:** PLA; GA; electrospinning; multilayer films; bioactive packaging

## 1. Introduction

The development of novel "functional foods" is a great opportunity to improve the quality of foods available to consumers to benefit their health and well-being in most modern societies. Indeed, the European Union (EU) is fighting diseases characteristic of a modern age, such as obesity, osteoporosis, cancer, diabetes, allergies, and dental problems [1]. A food can be regarded as functional if, beyond its inherent nutritional effects, it does satisfactorily demonstrate to provide added physiologic benefits (e.g., reduction of chronic disease risk) [2]. Currently, the majority of commercial functional foods are presented with the bioactive components contained within compatible foods. However, this aspect imposes to the food industry a number of limitations and difficulties during processing and storage such as the loss of product functionality, the development of undesirable flavors, or the modification of the food product texture [3]. The functional, or more precisely, "bioactive packaging" concept, is based on the incorporation into the packaging structure of the desired bioactive principles in optimum conditions until their eventual release into the food product either through controlled or fast release during storage, or just before consumption, taking into account the specific product/functional substance characteristics or requirements [4]. Indeed, the so-called bioactive packaging differs from the well-known active packaging technologies in the fact that while active packaging primarily deals with maintaining or increasing quality and safety of packaged foods (e.g., antimicrobial and antioxidant packaging), bioactive packaging provides a direct impact on the health of the consumer by generating healthier packaged foods.

Biodegradable polymers and sustainable plastics are nowadays essential elements involved in novel bioactive packaging technologies. Biopolymers are very advantageous to develop compostable (including edible) active and bioactive films due to their availability, non-toxicity, biodegradability and renewability, and their unique properties [5]. Biodegradable and/or sustainable polymer materials, such as synthetic biodegradable polymers, bio-based polyesters and polyamides, and naturally occurring polymers (e.g., proteins and polysaccharides and derivatives) are considered among the most suitable materials for the controlled release of bioactive substances [6]. Biopolymers will also act as an "added value" argument within the food chain for the upward use of sustainable packaging. In this sense, polylactide (PLA) is a commercially available bio-based and biodegradable polymer, which is gaining acceptance for both existing and novel applications in the packaging field in the form of different articles such as films or injection-molded pieces and parts [7]. Creating a fully based PLA film with control release ability of bioactives is certainly a new concept of sustainable functional packaging. Additionally, PLA is also a Food and Drug Administration (FDA) approved material because it is degraded by hydrolysis to products that can be metabolized and excreted [8].

In this context, gallic acid (GA), that is, 3,4,5-trihydroxybenzoic acid, is a naturally occurring polyphenol commonly found in a variety of fruits and vegetables, such as grapes, cherry, tea leaves, and longan seeds [9], either in its free or bound form (e.g., gallotannins) [10]. Moreover, GA can be obtained from both liquid and solid wastes of the agro-food industry, for instance the wine industry [11], resulting in a good candidate of a waste material converted into an added-value product. GA has been shown to exhibit bioactive properties such as antioxidant, anti-inflammatory, anticarcinogenic, and antifungal properties [12]. Nevertheless, GA is unstable at high temperatures or in the presence of oxygen or light, conditions that are common in food processing and storage [13]. Due to these factors, there is a very large trend to encapsulate and control the release of GA to improve its stability and bioactivity over time [14]. To this end, the composition of the encapsulating material is a major determinant of both the capsule's functional properties and the way it can be applied to improve the performance and release of a particular bioactive compound [15].

Novel methods are, therefore, pursued in a particular effort to incorporate GA in sustainable packaging materials avoiding losses in antioxidant activity during film formation or packaging structure development. In this regards, the electrohydrodynamic processing (EHDP), also called electrospinning when fibrilar structures are produced, has been recently suggested as being a simple and straightforward method of generating nanostructured encapsulation structures for a variety of

bioactive agents [16]. The electrospinning technique makes use of electrostatic forces to produce electrically charged jets from viscoelastic polymer solutions that upon drying, by the evaporation of the solvent, give rise to submicro- and nano-sized polymer-based fibers [17]. In this sense, EHDP is a promising technology that can be easily performed at room temperature so that it has recently attracted increased interest both for the nanoencapsulation and stabilization of thermolabile and light-sensitive substances [18–20] and for the development of fiber-based active coatings [21–24]. As a result, various bioactive-loaded electrospun biopolymer nanofibrous mats have been recently intended for food biopackaging purposes [25–27]. Moreover, in a more packaging application context, the electrospun fiber mats can be further converted into continuous films by the application of a thermal post-treatment below the biopolymer's melting point, the so-called annealing [28–30].

According to the above, the aim of this research work is to originally develop innovative multilayer structures of PLA with controlled release capacity of antioxidant GA that could serve as bioactive films for food and pharmaceutical packaging. To develop the bilayer structures, the electrospun PLA fibers containing GA were applied in the form of coating on PLA films previously prepared by cast extrusion. For the multilayers, a cast film of PLA was, first, coated on one side with different amounts of electrospun GA-loaded PLA fibers and then sandwiched with another PLA film on the other side. Thereafter, the three layers were annealed to generate a continuous film of PLA containing GA in the interlayer. The morphology, transparency, thermal properties, release profile, and antioxidant activity of the resultant PLA bi- and multilayer films were evaluated to ascertain their potential in bioactive packaging applications.

## 2. Materials and Methods

### 2.1. Materials

Ingeo™ 6201D grade, provided by NatureWorks (Minnetonka, MN, USA), was used for electrospinning. This is a fiber-grade PLA resin, derived primarily from annually renewable resources and supplied in pellet form. It has a density of 1.24 g/cm$^3$ and a met flow rate (MFR) of 15–30 g/10 min (210 °C and 2.16 kg). For the manufacture of the films, a PLA Ingeo™ 2003D grade also supplied by NatureWorks was used. This is general-purpose extrusion-grade PLA resin with a density of 1.24 g/cm$^3$ and a MFR of 6 g/10 min (210 °C and 2.16 kg).

Gallic acid (GA) with commercial reference G7384, having 97.5–102.5% (titration) and 170.12 g/mol, was supplied in powder form by Sigma-Aldrich S.A. (Madrid, Spain). This is a water-soluble phenolic acid obtained from grapes and the leaves of different plants.

Dichloromethane (DCM), N,N-dimethylformamide (DMF), and methanol, all with 99.8% purity, were supplied by Sigma-Aldrich S.A. The agent 2,2-diphenyl-1-picrylhydrazyl (DPPH), with a 394.32 g/mol, was also obtained from Sigma-Aldrich S.A.

### 2.2. Film Extrusion

Initially, all PLA pellets were dried at 60 °C for 24 h. PLA films with an average thickness of approximately 10 μm and 200 μm were obtained in a cast-roll machine MINI CAST 25 from EUR.EX.MA. (Venegono, Italy). The temperature profile was set at 195 (feeding)–195–200–200–205–210–210 (head) °C for both thicknesses. The extrusion speed was set at 30 rpm and 48 rpm for the 10-μm and 200-μm films, respectively. The speed of the calendar and the drag was adjusted to obtain the thickness requested in each case. The resultant films were annealed at 70 °C to develop crystallinity and release the remaining stresses in a vacuum drying oven Vaciotem-TV from S.P. Selecta S.A. (Barcelona, Spain). As a result, the PLA films achieved a completely smooth and flat surface. Finally, the films were then cut using a die on a hydraulic press model MEGA KCK-15A from Melchor Gabilondo S.A. (Vizcaya, Spain) to obtain square samples of 15 × 15 cm$^2$ [31].

### 2.3. Solution Preparation

A PLA solution was prepared for electrospinning by dissolving 10% ($w/v$) of the biopolymer in a DCM/DMF 7:3 ($v/v$) mixture at room temperature. GA was then added to the solution at a fixed content of 40 wt% based on the PLA weight. The resultant solution was gently stirred for up to 8 h at room temperature until a homogenous solution was obtained.

### 2.4. Solution Characterization

Prior to electrospinning, all PLA solutions were characterized in terms of surface tension, conductivity, and viscosity. Surface tension was measured following the Wilhemy plate method using an EasyDyne K20 tensiometer from Krüss GmbH (Hamburg, Germany). Conductivity was evaluated using a conductivity meter XS Con6 from Lab-box (Barcelona, Spain). Apparent viscosity ($\eta a$) was determined at 100 s$^{-1}$ using a rotational viscosity meter Visco BasicPlus L from Fungilab S.A. (San Feliu de Llobregat, Spain) equipped with a low viscosity adapter (LCP). All measurements were performed in triplicate at room temperature.

### 2.5. Electrospinning

Electrospinning was performed using a Fluidnatek® LE-50 benchtop line with temperature and relative humidity (RH) control system from Bioinicia S.L. (Valencia, Spain) with a variable high-voltage 0–30 kV power supply. This device was equipped with a motorized injector able to scan towards a metallic collector to obtain a homogeneous electrospun deposition. The PLA solution was first transferred to a 60-mL plastic syringe, which was connected through polytetrafluoroethylene (PTFE) tubes to a stainless-steel needle (Ø = 0.9 mm) whereas the needle tip was connected to the power supply. A 200-μm film of PLA was placed on the collector and the PLA solution containing GA was electrospun for 1, 2, and 3 h under a steady flow-rate of 2.2 mL/h using the motorized injector. The distance between the injector and collector was optimal at 18 cm and the voltage was set at 18 kV. PLA fibers without GA were also electrospun for 1 h in identical conditions as the control material. The process was carried out at 25 °C and 40% RH.

### 2.6. Multilayer Preparation

The PLA films one side coated with the electrospun PLA fibers were first sandwiched with a 10-μm PLA film and then subjected to thermal post-treatment in a 4122-model press from Carver, Inc. (Wabash, IN, USA). This process was performed at 150 °C, without pressure, for 120 ± 1 s. The resultant films were air cooled at room temperature for 240 ± 1 s. Prior to annealing, the electrospun mats were equilibrated for, at least, 1 week in a desiccator at 25 °C and 0% RH by using silica gel.

Before testing, the thickness of films was measured using a digital micrometer (S00014, Mitutoyo, Corp., Kawasaki, Japan) with ± 0.001 mm accuracy. Measurements were performed and averaged in five different points, two in each corner and one in the middle. Table 1 shows the codification and structure of the different prepared films with their corresponding thicknesses.

**Table 1.** Code and total thickness of the polylactide (PLA) films containing gallic acid (GA) according to their structure and deposition time during electrospinning.

| Film | Structure | Time (h) | Thickness (μm) |
|---|---|---|---|
| Monolayer | 200-μm PLA | 0 | 200.1 ± 0.9 |
| Bilayer 1h | 200-μm PLA/Electrospun PLA+GA fibers | 1 | 210.2 ± 2.8 |
| Bilayer 2h | 200-μm PLA/Electrospun PLA+GA fibers | 2 | 215.0 ± 3.1 |
| Bilayer 3h | 200-μm PLA/Electrospun PLA+GA fibers | 3 | 225.2 ± 2.0 |
| Multilayer 1h | 200-μm PLA/Electrospun PLA+GA film/10-μm PLA | 1 | 213.3 ± 1.8 |
| Multilayer 2h | 200-μm PLA/Electrospun PLA+GA film/10-μm PLA | 2 | 214.8 ± 1.9 |
| Multilayer 3h | 200-μm PLA/Electrospun PLA+GA film/10-μm PLA | 3 | 220.1 ± 2.1 |

*2.7. Material Characterization*

2.7.1. Morphology

The morphology of the electrospun fibers and the cross-sections of the films were observed by field emission scanning electron microscopy (FESEM) in a ZEISS ULTRA 55 from Oxford Instruments (Abingdon, United Kingdom). Film specimens were cryo-fractured by immersion in liquid nitrogen and then mounted on aluminum stubs perpendicularly to their surface. The working distance (WD) varied in the 6–7 mm range and an extra high tension (EHT) of 2 kV was applied to the electron beam. Due to their non-conducting nature, the films were subjected to a sputtering process with a gold-palladium alloy in a sputter coater EMITECH-SC7620 from Quorum Technologies, Ltd. (East Sussex, United Kingdom). The average fiber diameter was determined via ImageJ Launcher v 1.41 software using, at least, 20 FESEM images.

2.7.2. Thermal Analysis

Thermal stability was determined by thermogravimetric analysis (TGA) in a Mettler-Toledo TGA/SDTA 851 thermobalance (Schwerzenbach, Switzerland). Samples with an average weight between 5 and 7 mg were placed in standard alumina crucibles of 70 μL and subjected to a heating program from 30 °C to 700 °C at a heating rate of 20 °C/min in air atmosphere. The first derivative thermogravimetry (DTG) curves were also determined, expressing the weight loss rate as the function of time. All tests were carried out in triplicate.

2.7.3. Release Measurements

The release of GA was determined in saline medium following the procedure described by Chuysinuan et al. [32]. This type of medium was selected due to both the improved water retention behavior of the electrospun PLA mats and the high release rates of GA achieved. For this, the bi- and multilayer films containing GA were initially cut in square pieces of $20 \times 20$ mm$^2$ in randomly selected areas. The films were weighted and submerged in 30 mL of saline medium, prepared by dissolving 9 g of NaCl in 1 L of distilled water and stored for a whole period of ten weeks, that is, 1680 h, without darkness and at room temperature, that is, $23 + 2$ °C. At different submersion times, 1 mL of the medium solution was withdrawn (hereafter, a sample solution) and an equal amount of the fresh medium was refilled. The amount of GA in the sample solutions was determined in an Ultraviolet-Visible (UV-VIS) UV 4000 spectrophotometer (Thermo Scientific, Waltham, MA, USA) at a wavelength of 259 nm. The obtained data were calculated to determine the amount of GA released from the specimens at each submersion time point. For this, a calibration curve of GA in normal saline was previously determined, resulting in the following regression equation $Y = 0.0227X + 0.0156$ ($R^2 = 0.9989$), where "Y" is the absorbance and "X" the GA content, in ppm, of the sample. All measurements were carried out in triplicate.

2.7.4. Antioxidant Activity

The antioxidant activity of the GA released from the PLA films was determined by the DPPH inhibition assay. Measurements were taken after 2, 6, and 12 weeks of submersion in the saline medium. To this end, a modification of the procedure carried out by Chuysinuan et al. [32] was followed. Briefly, an aliquot of 1 mL of the sample solution was diluted with 5 mL methanol. Later, 1 mL of the resultant solution was mixed with 3 mL of $9.4 \times 10^{-2}$ M DPPH solution and the resulting solution was incubated for 30 min at room temperature in darkness. The absorbance of the final solution was recorded at the wavelength of 517 nm in the UV-VIS UV 4000 spectrophotometer. The antioxidant capacity of the extract was expressed as a percentage of inhibition of DPPH radical (% inhibition of DPPH radical) using the following equation:

$$I(\%) = \frac{A_0 - A_1}{A_0} \cdot 100 \tag{1}$$

where $A_0$ is the absorbance of the control and $A_1$ is the absorbance of the extract/standard. All measurements performed in triplicate.

### 2.7.5. Statistical Analysis

The solution properties, release of GA, and the antioxidant activity were evaluated through analysis of variance (ANOVA) using STATGRAPHICS Centurion XVI v 16.1.03 from StatPoint Technologies, Inc. (Warrenton, VA, USA). Fisher's least significant difference (LSD) was used at the 95% confidence level ($p < 0.05$). Mean values and standard deviations were also calculated.

## 3. Results

### 3.1. Solution Properties

Prior to electrospinning, the properties of the neat PLA and the GA-containing PLA solutions were measured. Table 2 shows the values of surface tension, conductivity, and viscosity. The determination of these parameters is certainly relevant to explain the behavior of each biopolymer solution during electrospinning and a combined optimization of them is needed to tune the fiber into the required form [19]. As one can observe, the addition of GA to the PLA solution mainly induced an increase in conductivity as well as a slight decrease in viscosity and a negligible effect on the surface tension. On the one hand, the conductivity increase observed when GA was dissolved in the biopolymer solution can be ascribed to ionic factors that induced a greater charge carrying capacity. A moderate increase in conductivity is preferred in PLA solutions since fiber formation habitually improves by an increase of surface charge of the spinning jet [33]. Indeed, if the conductivity of the polymer solution is relatively high, the force required for a polymer droplet to fall on the metallic target will be comparatively lesser and, thus, fiber formation will be enhanced. In other words, the fiber jet of higher conductive solutions will be subjected to a greater tensile force in the presence of a given electric field. On the other hand, the slight reduction attained in viscosity suggests that the presence of GA also induces a plasticizing effect on PLA.

**Table 2.** Values of surface tension, conductivity, and viscosity of the neat polylactide (PLA) and PLA containing gallic acid (GA) solutions.

| Solution | Surface Tension (mN/m) | Conductivity (µS/cm) | Viscosity (cP) |
|----------|------------------------|----------------------|----------------|
| **PLA** | 30.6 ± 0.1 [a] | 1.52 ± 0.08 [b] | 204.4 ± 0.4 [c] |
| **PLA + GA** | 30.7 ± 0.1 [a] | 5.83 ± 0.11 [b] | 195.9 ± 0.7 [c] |

[a–c] Different letters in the same column indicate a significant difference ($p < 0.05$).

### 3.2. Morphology

Figure 1 shows the morphology of the electrospun mats obtained from the FESEM analysis. One can observe in Figure 1a that the neat PLA solution yielded smooth and homogenous fibers with a mean diameter of 1.1 ± 0.3 µm. Similar fibers, though slightly thinner, were obtained after the incorporation of GA. As shown in Figure 1b, the GA-containing PLA fibers presented a mean diameter of 0.9 ± 0.2 µm. In both cases, the electrospun PLA fibers were homogeneous and fully free of beads or beaded regions. The study of Aytac et al. [34] describing the preparation of electrospun PLA containing GA also reported a diameter decrease in the fibers when the bioactive was encapsulated. This previously observed morphological change was related to both a viscosity increase and a conductivity decrease due to the presence of GA in the PLA solution. As the conductivity of the solution increases, the diameter of the submicron fibers decreases owing to the increment in the number of charges leading greater stretching of the polymer jet [35]. Another relevant observation produced after the incorporation of GA was the change of fibers surface from smooth to a rough or nearly ribbon-like shape. This morphological change can be related to an increase of the electrostatic

repulsion between charges encountered on the jet surface during electrospinning of the solution containing GA, which induced the formation of slightly thinner PLA fibers but also impaired the formation of a continuous surface along the fibers.

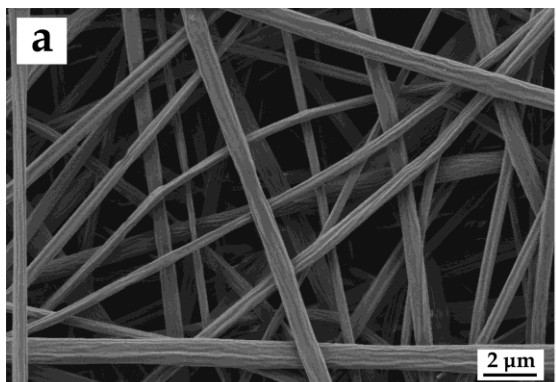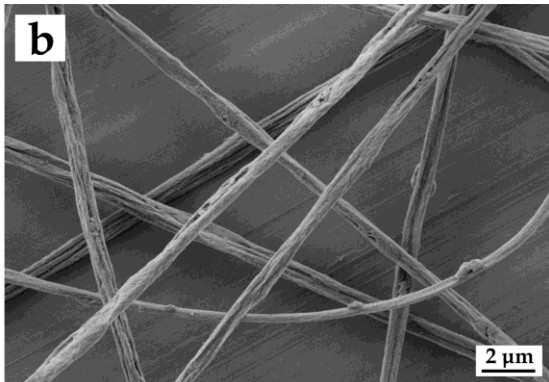

**Figure 1.** Field emission scanning electron microscopy (FESEM) micrographs of the electrospun fibers of: (**a**) Neat polylactide (PLA); and (**b**) PLA containing gallic acid (GA). Both images were taken at 5000× and scale markers of 2 μm.

In Figure 2 one can observe the FESEM images of the cross-sections of the PLA films prepared by cryo-fracture. Figure 2a shows the neat PLA film, which presented a smooth surface fracture that is representative of a brittle material. Figure 2b–d correspond to the bilayer films obtained after one, two, and three hours of electrospinning deposition of the GA-containing PLA fibers, respectively, without any thermal post-treatment. In these images one can clearly observe the presence of electrospun coatings of PLA fibers on the PLA films. As expected, the thickness of the electrospun coatings increased as a function of the deposition time during the electrospinning process. Thus, electrospun layers of PLA fibers of approximately 60, 85, and 100 μm were respectively obtained for the bilayers after one, two, and three hours of electrospinning. Although the electrospinning process yielded the formation of a continuous coating, one can also observe that it lacked of adhesion and a gap between the PLA film and the electrospun mat was observed, which most likely produced during cryo-fracture. Similar observations were recently described by Cherpinski et al. [36] where different biopolymer electrospun mats were not strongly adhered to a paper substrate. On the contrary, in the case of the multilayers, shown in Figure 2e–g, a continuous film structure without gaps was attained after annealing. In the case of the multilayers produced for 1 and 2 h of electrospinning deposition, the so-called multilayer 1 h and multilayer 2 h samples, respectively shown in Figure 2e,f, the electrospun interlayers were extremely thin in comparison to the overall multilayer thickness and then they were difficult to discern. In the case of the multilayer obtained after three hours of electrospinning deposition, that is, the so-called multilayer 3 h, shown in Figure 2g, a nanostructured interlayer with a total thickness of about 15 μm can be seen in the top region of the PLA film. The electrospun interlayers of GA-containing PLA also showed very little porosity, in the nanometric range, which can be ascribed to the large surface-to-volume ratio of the electrospun submicron fibers that efficiently coalesced and strongly adhered to both PLA films during the thermal post-treatment. In the multilayer 3h the fibrilar morphology was still kept in the electrospun interlayer, which can be mainly ascribed to an insufficient heat transmission due to the higher thickness of the electrospun interlay.

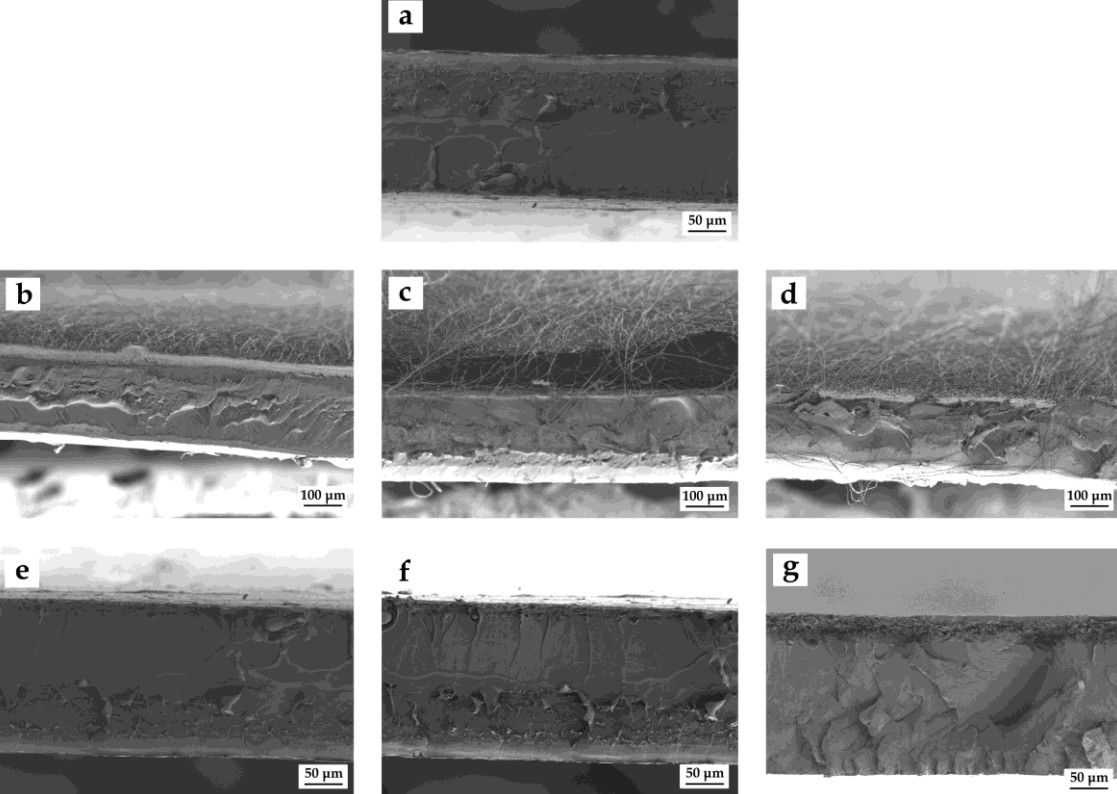

**Figure 2.** Field emission scanning electron microscopy (FESEM) micrographs of the gallic acid (GA)-containing polylactide (PLA) films of: (**a**) monolayer; (**b**) bilayer 1 h; c) bilayer 2 h; (**d**) bilayer 3 h; (**e**) multilayer 1 h; (**f**) multilayer 2 h; (**g**) multilayer 3 h.

*3.3. Film Transparency*

Figure 3 shows the optical appearance of the resultant PLA bi- and multilayers as a function of the deposition time during electrospinning. Simple naked eye examination of this image indicates that in the case of the bilayer structures, shown in the top images, the PLA films were mainly opaque. This effect is due to the inherent opacity of the electrospun coating, which is composed of fibers placed randomly that generate a significant level of porosity and hence refract the light very strongly [28,37]. Oppositely, the PLA multilayers, shown in the bottom images, highly retained the intrinsic transparency of PLA for up to 2 h of electrospinning. This optical effect can be related to the thermal post-treatment performed at 150 °C, below the biopolymer's $T_m$, the so-called annealing. This process effectively changed the electrospun interlayer morphology from a fiber-based mat structure to a continuous nanostructured film and hence reduced the porosity of the electrospun interlayer. During annealing a compact packing rearrangement of the electrospun nanofibers is produced by a phenomenon of coalescence [28,29]. For the multilayer 3h sample, however, the PLA film developed certain opacity due to the longer electrospinning deposition time. This effect can be ascribed to the high porosity of the electrospun interlayer since this film sample still retained partially the original fiber morphology, as previously shown in Figure 2g. From an application viewpoint, light penetration prevention (especially in the UV region) can also help reducing photoxidation of GA or any organic compounds present in foods. In any case, the annealing step applied on the PLA multilayers containing the electrospun interlayers resulted in continuous films with at least contact transparency, which have significant potential for use in food packaging applications [38]. This result indicates that the here-obtained nanostructured GA-containing fiber interlayers can be incorporated into PLA and the whole structure turned into actual films, which may be advantageous for encapsulation purposes and bioactive packaging applications.

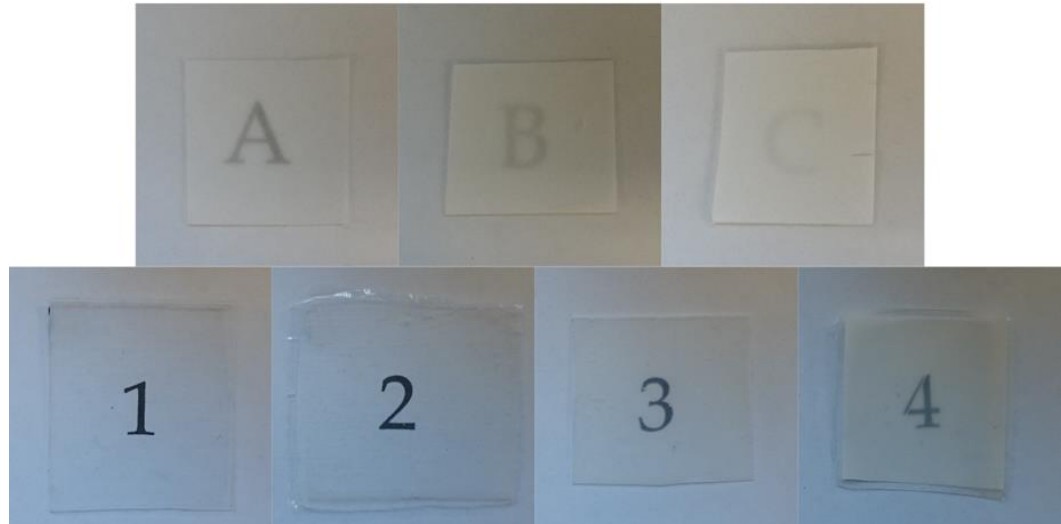

**Figure 3.** Visual appearance and contact transparency of the gallic acid (GA)-containing polylactide (PLA) films of: (**A**) bilayer 1 h; (**B**) bilayer 2 h; (**C**) bilayer 3 h; (**1**) monolayer; (**2**) multilayer 1 h; (**3**) multilayer 2 h; and (**4**) multilayer 3 h.

### 3.4. Thermal Stability

Figure 4 shows the TGA curves of the neat GA powder and the PLA films, whereas Table 3 summarizes the values obtained from these curves. On the one hand, it can be seen that the thermal decomposition of GA presented four peaks. The first mass loss occurred in the 70–130 °C range, which has been attributed to the degradation of volatile compounds of low molecular weight [39] as well as to the mass-loss event attributed to the water content [40]. The second and third peaks were centered at ca. 291 °C and 338 °C, respectively, which correspond to the main chain scission of GA and the loss of hydroxyl groups (–OH) [41]. Finally, minor and progressive degradation phenomena of lower intensity were observed at temperatures above 400 °C, which are related to the residual decomposition of GA at high temperature [39,40]. On the other hand, one can observe that all PLA films presented a practically identical pattern of mass loss, similar to that of the neat PLA film. In particular, PLA decomposed in a single step, starting at ~340 °C, with a main degradation temperature ($T_{deg}$) slightly above 375 °C [42]. The incorporation of GA provided a thermal stability improvement, even at the lowest GA content, due to the intrinsic antioxidant activity of the polyphenol. The onset degradation temperature, measured at the 5% of mass loss ($T_{5\%}$), was delayed for up to 10 °C, whereas the values of $T_{deg}$ also increased up to 5 °C. It is also worthy to mention that the thermal stability of the bilayer films was slightly better than that observed for the multilayer film, indicating that the capacity of GA to counteract the thermal effect was more effective when it was incorporated onto the film surface. For all film samples, the residual mass was nearly zero.

**Table 3.** Thermal degradation properties in terms of the onset degradation temperature ($T_{5\%}$), degradation temperature ($T_{deg}$), and residual mass at 650 °C of gallic acid (GA) and the GA-containing polylactide (PLA) films.

| Film | $T_{5\%}$ (°C) | $T_{deg}$ (°C) | Residual Mass (%) |
|---|---|---|---|
| GA | 261.5 ± 1.1 | 290.6 ± 1.1 / 337.5 ± 1.2 | 5.32 ± 0.12 |
| Monolayer | 338.1 ± 1.3 | 375.1 ± 1.4 | 0.12 ± 0.04 |
| Bilayer 1h | 348.2 ± 1.2 | 375.2 ± 1.1 | 0.11 ± 0.05 |
| Bilayer 2h | 348.9 ± 1.0 | 378.7 ± 1.2 | 0.12 ± 0.03 |
| Bilayer 3h | 349.1 ± 1.4 | 379.5 ± 1.3 | 0.19 ± 0.04 |
| Multilayer 1h | 345.3 ± 0.9 | 375.1 ± 1.1 | 0.14 ± 0.03 |
| Multilayer 2h | 346.4 ± 1.2 | 377.4 ± 1.0 | 0.20 ± 0.05 |
| Multilayer 3h | 347.3 ± 1.4 | 377.5 ± 1.2 | 0.18 ± 0.04 |

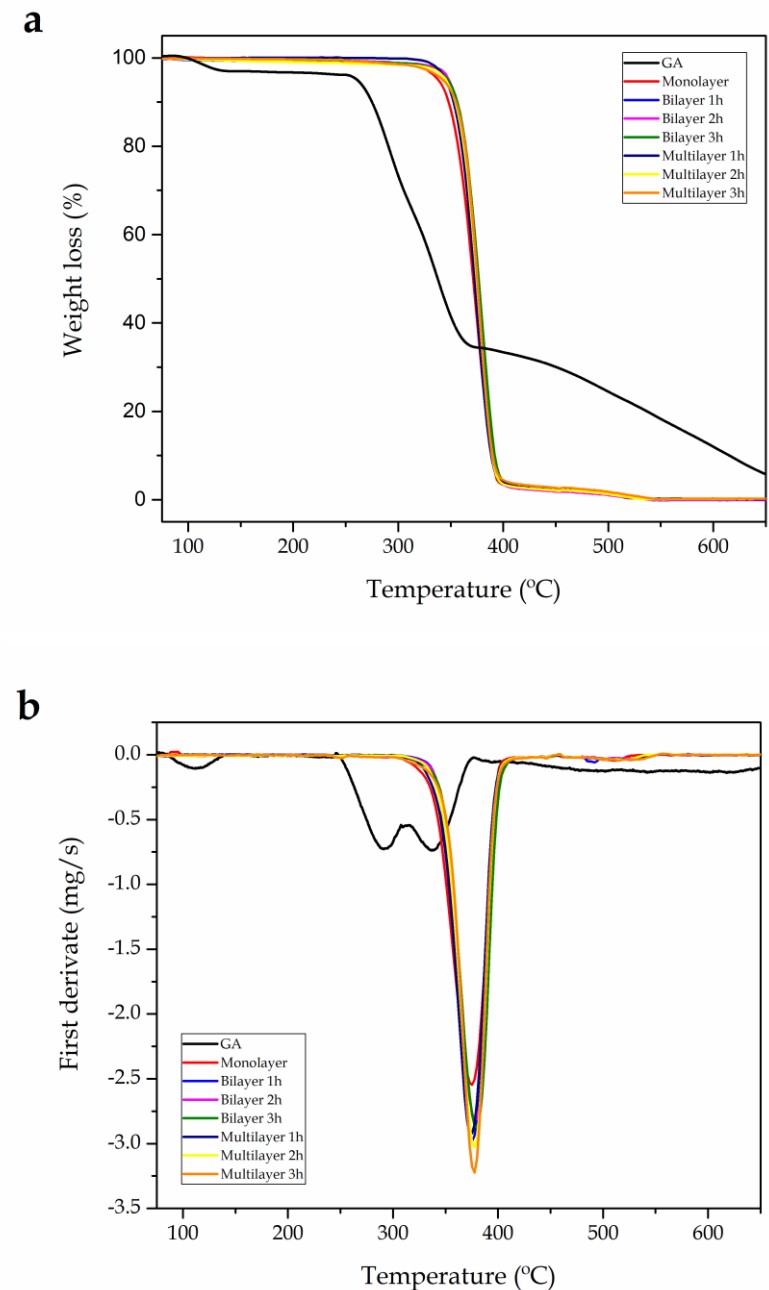

**Figure 4.** Comparative plot of the gallic acid (GA)-containing polylactide (PLA) films in terms of:
(**a**) Thermogravimetric analysis (TGA) curves; and (**b**) first derivative thermogravimetric (DTG) curves.

A similar behavior has been previously observed for other GA-containing films. For instance, Luzi et al. [41] recently described that the addition of 5 wt% GA delayed thermal degradation of poly(ethylene-*co*-vinyl alcohol) (EVOH) for up to about 20 °C respect to the neat EVOH film. In another study, Wu et al. [43] developed an active film based on gelatin with green tea extract (GTE), which contained 294.59 mg GA/g. It was observed that the thermal stability of the GTE-containing gelatin films was higher than that of the neat gelatin film and improved with increasing the concentration of GTE in the 0.3–0.7 wt% range. The increase of GTE did not only cause the $T_{deg}$ values of the gelatin films to increase but also decreased the maximum rate of mass loss. It is considered that interactions between the terminal –OH groups of the biopolymer chain with –OH, –CO, and/or –COOH chemical groups of the GA components via hydrogen bonding are responsible for the observed thermal stability improvement.

*3.5. In vitro Release Studies*

Figure 5 shows the GA release from the bi- and multilayer PLA films as a function of the immersion time in saline medium for a span time of 1680 h. The in vitro release kinetics data are depicted in terms of the total amount release of GA (Figure 5a) and its cumulative release to the medium based on the theoretical content of GA in the PLA films (Figure 5b). One can observe that the release profile of the PLA films considerably varied according to their structure and, as one could expect, the total amount of GA released to the medium increased with increasing the GA content in the film samples. As a general observation, all bilayer films showed a Fickean diffusion release mechanism. During the first 5 h, a burst release profile was attained, showing cumulative release values of approximately 83%, 46%, and 35% for the PLA films coated by electrospun mats for 1 h, 2 h, and 3 h, respectively. Upon immersion in saline media, it is considered that the PLA fibers' coating is rapidly hydrated and swells, thus allowing the dissolution of the embedded polyphenols and their subsequent release at a relatively high rate due to the highly porosity of the electrospun mat. It is worthy to note that this phenomenon of burst release was observed even though the coated films were immersed in a water-based medium, for which PLA presents a low chemical affinity. It can be considered that the Fickean diffusion process was overlapped by the swelling behavior of the electrospun PLA fibers, leading to high release rates and thus accelerating the process. One can also observe that the GA release rate was higher in the bilayer films processed for shorter electrospinning deposition times, which can be related to the lower thickness of the electrospun coating and, thus, to a more favored diffusion of the polyphenol to the saline medium. In a second step, the release rates progressively decreased for all PLA films coated at different electrospinning deposition times. For the bilayer film obtained by 1 h of electrospinning deposition, the GA release rate stabilized after only ~45 h. In the PLA films coated for 2 and 3 h, stabilization was attained after approximately 250 h. At the end of the experiment, all bilayer films completed a theoretical cumulative release of GA in the 82–92% range. The deviation from the ideal value of 100% can be related to the inhomogeneous distribution of GA in different parts of the obtained electrospun PLA layers, partial losses during processing, and also a minor accumulation of GA in the PLA fibers during the whole tested immersion time.

The above-described short or short-to-medium release can be of interest for most food packaging materials, based on the fact that the mean shelf life of a food-packaging product is typically below two weeks [44]. In this sense, Chuysinuan et al. [32] also studied the release behavior of GA from a single electrospun mat of PLA fibers in saline medium for 48 h. A similar release profile was reported, showing that the cumulative amount of GA released rapidly increased during the initial immersion time. Specifically, at 30 min after immersion, the cumulative amount of GA released into the normal saline was 21%. Further increase in the immersion time resulted in a gradual increase in the cumulative amount up to a plateau value of approximately 90% was reached. This burst release was related to the high water retention behavior of the GA-loaded PLA fibers, as well as potential dissociation of the polyphenol into ionic species due to the difference between the pKa value of GA and the pH of the normal saline, that is, 4.41 vs. 7. Moreover, the dissociation of the carboxylic acid chain ends of PLA into carboxylate groups (and protons) could additionally help promote the diffusion of the gallate ions out of the electrospun fibers due to electrostatic repulsion. Similarly, Phiriyawirut and Phaechamud [45] reported the release of GA encapsulated at different rations into electrospun cellulose acetate (CA) nanofibers. It was observed that the release from 7.5–10 wt% GA-loaded electrospun CA fiber mats in phosphate buffer medium at room temperature was also rather rapid during the first 2 h, reaching cumulative release values of 35–45% after 100 h. The release profile reported was related to relative high content of GA, suggesting that the polyphenol was mainly located on the surface of electrospun fibers due to its limited solubility in the CA solution for electrospinning.

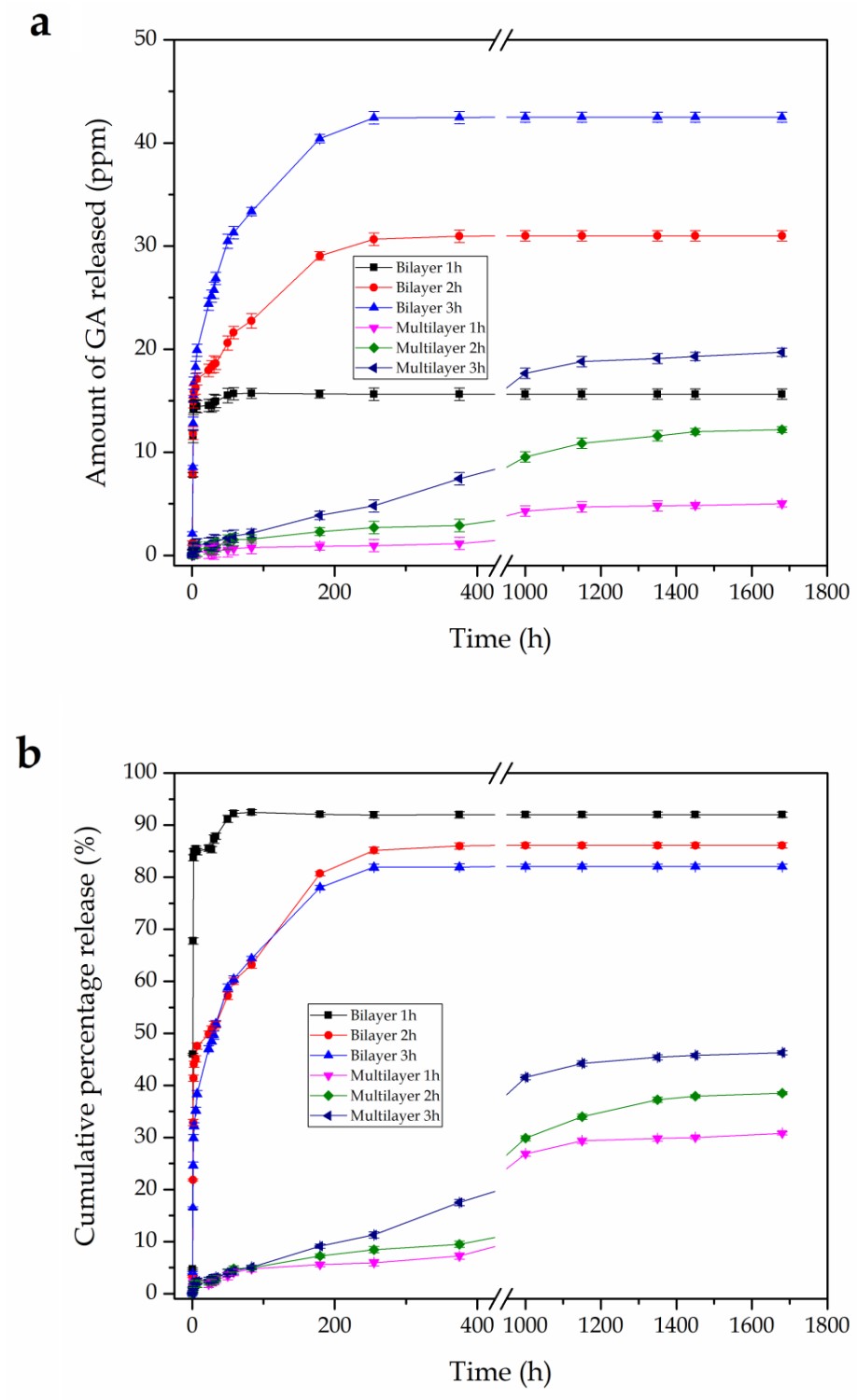

**Figure 5.** Gallic acid (GA) release from the GA-containing polylactide (PLA) films as a function to the immersion time in saline medium: (**a**) Total amount released; and (**b**) cumulative release.

Oppositely, the multilayer PLA films exhibited a highly sustained release profile of GA to the selected simulant, that is, the saline medium. It can be observed that the multilayer PLA films smoothly delivered the bioactive and also still retained high amounts of GA during the whole tested period, that is, 10 weeks. In particular, at 375 h, when the release rates of bilayer films were already stabilized, the cumulative GA release values were only 7.4%, 10.8%, and 17.3% for the PLA films based on interlayers electrospun for 1 h, 2 h, and 3 h, respectively. Then, the three multilayers started to stabilize

at approximately 1000 h, nearly reaching a plateau. The final cumulative GA values for the multilayers containing the electrospun interlayers processed for 1 h, 2 h, and 3 h were approximately 30%, 38%, and 46%, respectively. It can be then considered that the loaded GA was highly confined in the electrospun PLA interlayer due to the presence of the 10-μm top PLA layer. Therefore, although the release rate was stabilized, a slow and prolonged release is expected to occur for longer immersion times (no more data was collected). Since the GA release is the result of the diffusion and equilibrium process involving its transfer from the electrospun interlayer into the food simulant, the top PLA layer acted as an efficient barrier against the GA diffusion due to its intrinsic hydrophobic behavior. In this regard, it has been reported that the water absorption of PLA is only 0.7 wt% [46]. Therefore, the release mechanism in the fully PLA-based multilayer system can be easily further controlled and then tuned by not only adjusting the deposition time of the electrospun interlayer but also by selecting the appropriate thickness of the top layer prior to the thermal post-treatment.

### 3.6. Antioxidant Activity

Finally, the antioxidant capacity of the GA released from the PLA films was evaluated by the DPPH radical scavenging assay to assess their potential in bioactive packaging applications. DPPH is a stable radical with a maximum absorption at 517 nm that can readily undergo scavenging by an antioxidant such as polyphenols [47]. This method is based on the reduction of the relatively stable radical, that is, DPPH, to the formation of a non-radical form in the presence of hydrogen donating antioxidants. GA is a phenolic compound containing hydroxyl groups attached to the benzene ring in the ortho position so that it possesses electron and/or hydrogen donor ability [48]. Indeed, the antioxidant activity of these compounds is mainly due to their redox properties, which allows them to act as reducing agents, hydrogen donators, and singlet oxygen quenchers having metal–chelating capacities [49]. Table 4 shows the antioxidant capacity of the GA released from the bi- and multilayer PLA films after their immersion for two, six, and ten weeks in the saline medium.

**Table 4.** Inhibition (%) of 2,2-diphenyl-1-picrylhydrazyl radical (DPPH) of the gallic acid (GA) released from the polylactide (PLA) films.

| Film | Inhibition (%) | | |
|---|---|---|---|
| | **2 Weeks** | **6 Weeks** | **10 Weeks** |
| Bilayer 1h | 42.4 ± 0.7 [a] | 38.4 ± 0.9 [a] | 25.6 ± 1.2 [a] |
| Bilayer 2h | 65.7 ± 0.9 [a] | 54.9 ± 1.2 [a] | 47.7 ± 0.9 [a] |
| Bilayer 3h | 96.0 ± 2.4 [a] | 78.9 ± 1.3 [a] | 54.7 ± 1.3 [a] |
| Multilayer 1h | 5.9 ± 0.8 [b] | 29.2 ± 0.7 [b] | 38.1 ± 0.7 [b] |
| Multilayer 2h | 8.1 ± 0.8 [b] | 37.4 ± 1.0 [b] | 52.4 ± 1.2 [b] |
| Multilayer 3h | 12.3 ± 1.2 [b] | 48.7 ± 0.9 [b] | 71.4 ± 0.8 [b] |

[a–b] Different letters in the same column indicate a significant difference ($p < 0.05$).

As one can observe in the table above, the GA released from all the PLA films showed antioxidant activity confirming that electrospinning process was successful in retaining the bioactivity of GA [14,50], even after the electrospun mats had been subjected to a high electrical potential during the fiber producing process as well as the thermal post-treatment. The antioxidant activity of the GA released from the PLA films was in agreement with the total amounts of the as-released GA, previously shown in Figure 5a. Therefore, in the case of the bilayer PLA films, a strong antioxidant activity was attained after two weeks of immersion in the medium while it progressively decreased over time. On the contrary, the multilayer PLA films provided a release of GA that showed a relatively low DPPH inhibition in first weeks. However, after ten weeks, the PLA films containing the electrospun GA-loaded interlayer achieved a stronger DPPH inhibition, indicating that it effectively protects the polyphenol over longer periods. Based on the present observations, it can be considered that the GA release and, thus, the GA antioxidant functionality can be controlled by the appropriate design of PLA films containing electrospun coating and interlayers.

In this sense, other previous studies have shown the potential of electrospinning to encapsulate GA in order to retain its antioxidant activity for active packaging applications. In the study performed by Neo et al. [14], GA was successfully encapsulated at 5–20 wt% in zein nanofibers by electrospinning. It was found that the DPPH scavenging abilities of the GA-loaded zein electrospun fibres varied from 58% to 89%, showing that there was no significant difference for DPPH scavenging activity of GA before and after the electrospinning process. Similar antioxidant properties have been recently observed by Aydogdu et al. [27], who loaded GA at 2–10 wt% into electrospun hydroxypropyl methylcellulose (HPMC)/ polyethylene oxide (PEO) blend nanofibers. The antioxidant activity of the electrospun HPMC/PEO nanofibers considerably increased as the GA content increased, reaching an antioxidant activity of up to 24.74 mg DPPH/g dry weight and 50.35%. However, in both studies the electrospun nanofibers had first to be fully dissolved in ethanol aqueous solutions in order to effectively release the encapsulated GA and then achieve the reported antioxidant activity.

## 4. Discussion

In this study, homogenous ultrathin PLA fibers incorporating GA polyphenol at 40 wt% with a mean diameter of ca. 0.9 µm were successfully produced by the electrospinning technique. The incorporation of GA into the PLA fibers increased the roughness surface and also slightly flattened the fibers shape mainly due to the conductivity increase observed in the solutions used for electrospinning. The GA-containing submicron PLA fibers were then used to build nano-sized bi- and multilayers structures based on previously prepared cast-extruded 200-µm PLA films. For the bilayer structures, the electrospun mats were applied as a coating on the PLA films for 1, 2, and 3 h. Electrospun coatings of approximately 60, 85, and 100 µm were then obtained on the PLA films after 1, 2, and 3 h of deposition during electrospinning. The electrospinning process successfully yielded the formation of continuous coatings made of submicron PLA fibers, but they were not strongly adhered to the PLA film and the resultant bilayer PLA films also lacked of transparency. To produce the multilayer PLA films, the cast films of PLA were first one side coated with electrospun GA-loaded PLA fibers, also processed for 1, 2, and 3 h, and then sandwiched with a thin PLA film of 10 µm on the other side by the application of a thermal post-treatment without pressure, the so-called annealing. It was also observed that the thickness of the electrospun nanostructured interlayers increased as a function of the during the electrospinning process though they were relatively difficult to discern due to their low thicknesses. The resultant multilayer films interestingly presented sufficient adhesion between layers as a result of the large surface-to-volume ratio of the electrospun fibers that efficiently coalesced and adhered strongly to both PLA film sides during annealing. The multilayer PLA films showed relative high transparency when the containing nanostructured interlayers were produced for up to 2 h of electrospinning due to their very little porosity.

Results also showed that the incorporation of the natural polyphenol positively delayed the thermal degradation of PLA for up to 10 °C and all films were thermally stable up to 345 °C. This thermal enhancement was more pronounced in the case of the bilayer films, being this phenomenon ascribed to the presence of the antioxidant in regions close to the films surface. During the in vitro release studies, the GA release amounts from all PLA films increased but the release rates decreased with increasing the thickness of the electrospun GA-containing layer. The bilayer PLA films showed that the cumulative release amount of GA increased rather rapidly for the first 5 h of immersion time. Further increase in the submersion time resulted in a gradual increase in the cumulative amount of the polyphenol up to it finally reached a *plateau* value after 45–250 h. On the contrary, the PLA multilayers offered a highly sustained release of GA, having the capacity to release the polyphenol for over 1000 h and being the release diffusionally controlled. Therefore, whereas the bilayer PLA films can provide a quick release of antioxidant into saline food simulant that can be of interest for functional food in packaging materials of a relatively short life cycles, that is, 1–2 weeks, the multilayer systems can be interesting from a point of view of food products stored in water-based media with a much longer shelf life. Finally, the antioxidant activity assessment of the GA released from the PLA films was carried out

by measuring its capacity to scavenge the DPPH radicals. During the first days, the bilayer PLA films performed as a potent source of antioxidant while the multilayer films offered comparatively weaker activities. In the long term, however, the GA released from the multilayer PLA films were able to show a stronger DDPH inhibition.

This work has demonstrated that it is possible to develop multilayer PLA films containing electrospun nanostructured coatings and interlayers that provide specific release rates of the antioxidant GA into saline medium. The fabricated multilayer films connect the rising technology of bioactive packaging intended to aid in the production of functional foods, that is, to enhance the impact of food on the consumer's health. Further studies should be focused on studying the GA release profiles of the here-developed multilayers study in different solutions and also to apply them in real food products.

**Author Contributions:** Conceptualization was devised by R.B., J.M.L., and S.T.-G.; methodology, validation, and formal analysis was carried out by L.Q.-C., N.M., and S.T.-G.; investigation, resources, data curation, and writing—original draft preparation was performed by L.Q.-C. and S.T.-G.; writing—review and editing: L.Q.-C., R.B., and S.T.-G.; supervision: J.M.L., R.B., and S.T.-G; project administration: J.M.L. and R.B.

**Funding:** This research was supported by the Ministry of Science, Innovation, and Universities (MICIU) program numbers MAT2017-84909-C2-2-R and AGL2015-63855-C2-1-R and by the EU H2020 project YPACK (reference number 773872).

**Acknowledgments:** L.Q.-C. wants to thank Generalitat Valenciana (GV) for his FPI grant (ACIF/2016/182) and the Spanish Ministry of Education, Culture, and Sports (MECD) for his FPU grant (FPU15/03812). S.T.-G. acknowledges MICIU for his Juan de la Cierva–Incorporación contract (IJCI-2016-29675).

**Conflicts of Interest:** The authors declare no conflict of interest.

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
