# Peer review of "Bioactive Multilayer Polylactide Films with Controlled Release Capacity of Gallic Acid Accomplished by Incorporating Electrospun Nanostructured Coatings and Interlayers"

_applsci, doi:10.3390/app9030533_

Round 1

Reviewer 1 Report

This paper showed some merit and interest in that PLA cast nanofibre films were impregnated with Gallic Acid as an antioxidant.

The authors claim that a controlled release of gallic acid was demonstrated for use in storage and packaging materials of food. However, only a saline solution was used and real food matrices were not tested. Therefore I am unsure how this PLA material would react to high-fat food products especially as gallic acid is thermally unstable at high temperatures and under strong light conditions. I would have liked to have seen some controls applied to the experimental procedures, as without controls it is difficult for th authors to make any claims on their study.

The DPPH antioxidant assay was OK but can the authors justify why measurements were conducted after 14 immersion in saline?Figure 4 was difficult to determine so I would suggest changing the line colours for easy reading.

Lines 460-462: Stability in foods? This has not been determined - Conclusion should be rewritten for clarity.

References: It is usual convention to use Capitilisation on Journal names (see References 7; 16;17 and 22)

Full Journal name inserted into Reference 10

Reference 32 has no publisher?

Author Response

This paper showed some merit and interest in that PLA cast nanofibre films were impregnated with Gallic Acid as an antioxidant.

The authors claim that a controlled release of gallic acid was demonstrated for use in storage and packaging materials of food. However, only a saline solution was used and real food matrices were not tested. Therefore I am unsure how this PLA material would react to high-fat food products especially as gallic acid is thermally unstable at high temperatures and under strong light conditions. I would have liked to have seen some controls applied to the experimental procedures, as without controls it is difficult for th authors to make any claims on their study.

The objective of this study was to develop a first system with controlled release capacity, mainly aimed for food packaging though it can also be applied for other packaging applications (e.g., pharmaceutics). This medium was selected due to both the improved water retention behavior of PLA and the high release rates of gallic acid based on previous studies. The conclusion section was modified to better describe the application/potential of the materials developed and, as indicated by the reviewer, the future research activity in real food products was also addressed.

The DPPH antioxidant assay was OK but can the authors justify why measurements were conducted after 14 immersion in saline? Figure 4 was difficult to determine so I would suggest changing the line colours for easy reading.

There was a mistake in the experimental description: The antioxidant capacity of the bi- and multilayer PLA films was evaluated for immersion times of 2, 6, and 12 weeks in the saline medium. This has been corrected. Figure 4 was also modified to improve visualization.

Lines 460-462: Stability in foods? This has not been determined - Conclusion should be rewritten for clarity.

This sentence has been removed to avoid confusion and the conclusion section has been improved for clarity.

References: It is usual convention to use Capitilisation on Journal names (see References 7; 16;17 and 22)

All references currently match the MDPI journals style.

Full Journal name inserted into Reference 10

This reference has been amended.

Reference 32 has no publisher?

This reference has been amended.

Reviewer 2 Report

Dear Authors,

Congratulations on the completion of your study described in the manuscript “Bioactive Polylactide Multilayer Films with Controlled Release Capacity of Gallic Acid Accomplished by Incorporating Electrospun Nanostructured Coatings and Interlayers.” I have carefully read your draft paper and concluded that your study is useful and interesting and may be acceptable for publication after some minor revisions are successfully completed. I like and enjoyed reading your draft paper, but I have a few questions and concerns with your work as presented, which I invite the authors to address or explain, and which are detailed below in the file.

Please note that the comments are intended merely to assist the authors in improving the paper and ensuring that published papers are of the highest quality. They are in NO WAY intended to discourage or demean the authors personally.

Sincerely,

Reviewer.

Author Response

Abstract and discussion: Please consider a better clarification of your contribution with respect to other publications such as "...".

The abstract and discussion section has been improved to remark the contribution of the present study.

Introduction: Please expand to give a very clear and explicit statement of the goals and motivation for your study and state what is new and unique about your study.

A statement of the goals and motivation for the study has been added at the end of the Introduction section.

-          Materials and methods: this section is well explained and clear, however some details such as the number of tests and the deviation obtained in the results would be interesting in order to provide a more accurate assessment of them.

More information about the number of tests was added in each section.

-          Check the format of the document as there are slight errors such as in line 235 "One the one hand", a line break (488), and calls to figure or publication years in bold letters.

These mistakes have been amended.

-          A good explanation of the results is presented, although I comment certain details that I would like you to clarify during the text. For example, the morphological change of fibers surface from smooth to a rough or nearly ribbon-like shape, what improvement does it bring to the system?

The presence of gallic acid modified the morphology of the fibers, as indicated in the text, but it had no influence on the films and on their properties.

-          Figure 1: The increase in scale of the morphology of the surface does not provide clarification with respect to its equivalent on a smaller scale. Perhaps it should only be included on one scale, if you prefer the larger one.

As suggested, Figure 1c was removed.

-          Figure 2: The reader will appreciate the inclusion of images on the same scale, even if clarified. For example, Figure 2a is not appreciated in detail.

As requested, Figure 2a was replaced by a new one on a similar scale.

-          There is plenty of self-referral. These should only be used in some cases that are very necessary. Please check if you can substitute or avoid some of them as they may detract from the quality of the journal.

The number of self-references have been reduced and also new references from other authors have been included.

-          The conclusion/discussion section should be revised to summarize and emphasize the contributions of the publication, avoiding the repetition of some details of the methodology.

The conclusion section has been improved to emphasize the contribution of the study in the field.

Round 2

Reviewer 1 Report

The revised manuscript shows significant improvement. All of the suggested changes have been actioned by the authors and are satisfactory.

I still have a concern over Lines 803 - 806 and the statement made; 'Bioactive packaging intended to aid in the production of functional food ..... on consumer health'

This is an erroneous statement as bioactive packaging does not necessarily imply a connection with functional foods - it may help to extend shelf-life of food. Please review and rewrite the statement in the conclusion

Author Response

The revised manuscript shows significant improvement. All of the suggested changes have been actioned by the authors and are satisfactory.

I still have a concern over Lines 803 - 806 and the statement made; 'Bioactive packaging intended to aid in the production of functional food ..... on consumer health'

This is an erroneous statement as bioactive packaging does not necessarily imply a connection with functional foods - it may help to extend shelf-life of food. Please review and rewrite the statement in the conclusion

The "bioactive packaging" concept described here is based on the incorporation into the packaging structure of a bioactive principle (in this case gallic acid, a natural antioxidant) until their eventual release into the food product either through controlled (multilayer film) or fast (bilayer film) release during storage. Therefore, this type of packaging is not necessarily related to antioxidant (active) packaging, which would be aimed to extend shelf life of packaged food products, but it does have a direct association to the development of functional foods. The main difference between the well-known active packaging technologies and the bioactive packaging concept is that while active packaging primarily deals with maintaining or increasing quality and safety of packaged foods, bioactive packaging has a direct impact on the health of the consumer by generating healthier packaged foods. This information is supported in the manuscript by Reference 4 (“Bioactive packaging: turning foods into healthier foods through biomaterials”) and it has been added in the first paragraph of the introduction to avoid further confusion between the two packaging terms. Please see the new text, underlined in green, that complement the previous information already included in the original manuscript.